# A qualitative exploration of the experiences of pregnant women living with obesity and accessing antenatal care

Margaret Charnley[1], Lisa Newson[2], Andrew Weeks[3], Julie Abayomi[4]*

1 School of Health & Sort Sciences, Liverpool Hope University, Liverpool, United Kingdom, 2 School of Psychology, Faculty of Health, Liverpool John Moores University, Liverpool, United Kingdom, 3 Sanyu Research Unit, Department of Women's and Children's Health, University of Liverpool, Liverpool, United Kingdom, 4 School of Medicine and Nutrition, Health Research Institute, Faculty of Health, Social Care & Medicine, Edge Hill University, Ormskirk, United Kingdom

* abayomij@edgehill.ac.uk

**Data Availability Statement:** Ethical approval was not granted for the release of the full data transcripts (as this may compromise the privacy and identity of the individual research participants),

## Abstract

Women are advised to optimise weight before pregnancy. However, many are either already living with overweight or obesity prior to becoming pregnant, increasing the risks for adverse outcomes. Health care professionals (HCP) are responsible for advising women of risks throughout and following pregnancy. However, midwives often find broaching the conversation around maternal obesity difficult. This study explored the experiences of pregnant women living with obesity in accessing antenatal care. Seventeen women completed a semi-structured interview. Transcripts were analysed thematically. Four themes were developed: 1) antenatal care is inconsistent, 2) additional support is needed, 3) women feel judged about their weight, and 4) weight cycling is highly prevalent. Findings suggest that pregnant women living with obesity often experience weight bias from HCPs, feel judged because of their weight and are left feeling confused and overlooked. Women reported inconsistencies in advice and care offered and acknowledged a lack of continuity of care throughout pregnancy. We call for an urgent need for further multidisciplinary training to address the concerns, experiences and needs of pregnant women living with obesity.

## Introduction

Pregnancy is often the time when women gain the most weight in a relatively short time period [1]. There are no official UK guidelines relating to gestational weight gain (GWG). However, the USA Institute of Medicine (IOM) suggests limiting GWG to between 5 and 9kg for women with a body mass index (BMI) recorded in the obese category (BMI$\geq$30kg/m$^2$) for optimal pregnancy outcomes [2]. High pre-gravid BMI and excessive GWG are known predictors for immediate obstetric risk and long-term risk for obesity-related disease in both pregnant women and their offspring [3, 4]. Consequently, there is a growing emphasis on weight management and lifestyle interventions during and after pregnancy to address the associated risks.

but approval to publish extracts of data used as quotes for evidence to support analytical commentary and published within a journal article was approved, and participants explicitly provided informed consent for this level of release (please see quotes in the manuscript for examples of raw data). NHS Ethical approval was awarded via the Integrated Research Application System (IRAS), please contact service.desk@hra.nhs.ukemail for further clarification on ethical limitations. To discuss the data please contact the corresponding author by email [abayomij@edgehill.ac.uk].

**Funding:** MC - MerseyBEAT, Liverpool Primary Care Trust; Liverpool Health Inequalities Research Group (Ref MBRP005). The funders had no role in study design, data collection and analysis, decision to publish, or preparation of the manuscript.

**Competing interests:** The authors have declared that no competing interests exist.

In the UK, the Royal College of Obstetrics and Gynaecology [5] suggest that women should optimise their weight before pregnancy; they should receive advice about weight during pre-conception counselling and that women with a BMI $\geq$30 kg/m$^2$ should be advised about the risks of obesity during pregnancy and be supported to lose weight postpartum. The National Institute for Health and Care Excellence [6] also recommend achieving a healthy weight before conception and avoiding dieting whilst pregnant. There is limited guidance regarding what HCPs should advise during pregnancy to promote positive pregnancy outcomes. It is also well documented that health care professionals (HCPs), particularly midwives, are often reluctant to have conversations about weight with patients, as they feel that they lack sufficient knowledge and skills or they are concerned about offending the pregnant woman or having a negative impact on their professional relationship [7, 8].

The issue of obesity is complex and multi-faceted, with energy balance forming only one part of the story. There are socio-cultural and socio-economic factors as well a psycho-social experiences that influence food choice and behaviours, in addition to genetics and the environment, that may determine predisposition to obesity and obesity-related disease [9]. Society often responds negatively to obesity. Pregnant women living with obesity report experiencing weight stigma regularly (ranging from a few times per month to at least once a week), and two-thirds of 501 women described experiencing weight stigma from more than one source, with media being the most common source [10].

During antenatal care, women may be exposed to weight bias from HCPs, including from midwives. A recent scoping review [11] found that weight stigma occurred during antenatal care when HCPs avoided weight-related conversations, made assumptions about lifestyle behaviour and communicated risks associated with obesity poorly or insensitively. The authors recommended offering 'sensitivity training', enabling HCPs to discuss maternal obesity via a more patient-centred approach. There are also reports of poor quality antenatal care, worsening health behaviours and adverse pregnancy outcomes associated with weight stigma [12]. In the USA, Incollingo Rodriguez and colleagues [10] found that one in five women (n = 92) reported experiencing weight stigma during antenatal care, with obstetricians (33.8%) and nurses (11.3%) most likely to be the source [10]. Participants reported feeling judged (23.4%) and shamed (15.8%) because of their weight. These experiences undermined the patient-provider relationship, with 11.2% of participants reporting a lack of trust in their doctor and 7.7% changing healthcare providers due to their treatment. In a UK study, pregnant and postnatal women living with obesity who had declined referral to, or had disengaged with weight management services were interviewed [13]. Many women reported being upset on receiving the referral letter for weight management services; either because the referral had not been discussed with them (by the HCP who made the referral), or they did not receive enough information about it and so they felt that the service did not meet their needs. Further to this, some women expressed that HCPs would make assumptions about them and their ability to care for their child if they attended.

Understanding the experiences of living with obesity enables HCP to guide pregnant women towards more positive behaviour changes and to foster more meaningful discussions with regards to weight. However, perceptions of weight stigma, insensitivity, or feelings of being judged by HCPs may hinder such conversations. A more sensitive referral process may improve engagement with weight management services, with more details regarding what the service entails [13]. Some pregnant women living with obesity may not want access to weight management services, so HCPs should not make assumptions [13]. A study investigating GWG and dietary behaviour during pregnancy, though not focusing on those living with obesity, reported that although pregnant women acknowledged that healthy eating is important during pregnancy, women described weight gain as inevitable, so they were less inclined to

focus on weight during pregnancy [14]. Pregnant women felt motivated to make positive changes to their diets during pregnancy but reported insufficient knowledge and a lack of support from their HCPs to do so.

There is limited research focussing on the experiences of pregnant women in the UK who are living with obesity and seeking to understand their experiences of antenatal care. Therefore, this study explored the lived experience of pregnant women living with obesity. The objectives were to enhance our understanding of their journey through antenatal care and contribute valuable insights for shaping future guidelines managing obesity during pregnancy.

## Materials and methods

A qualitative method and analysis were employed for this research. The study used an inductive data-driven thematic analysis, as this provided a theoretically flexible approach that offered comprehensive narratives and exploration of patterned meaning across the entire data set [15]. Thematic analysis is used regularly in health research, as it is appropriate for applied research, such as maternal health [16]. This paper accounts for conversations with antenatal care service users (postnatal women living with obesity), who had previously participated in the quantitative arm of the Fit for Birth (FFB) study [17, 18]. Participants were between 12 and 24 months postnatal at the time of this study. Obesity is a complex health issue in which the individual living with obesity is blamed and where it is generally viewed as something that can be addressed via changes in lifestyle [19]. This study enabled the personal perspectives and experiences of women living with obesity during pregnancy to be better understood. Women living with obesity were interviewed to explore the impact of lifestyle and their lived experiences of antenatal care, to aid understanding of potential issues and to inform future guidelines for the care and management of obesity in pregnancy. A National Health Service ethics application was approved for this study (IRAS-ref number09/H1005/23).

### Participants

Semi-structured interviews were conducted with eighteen women who all expressed an interest in participating in this qualitative study during recruitment for the FFB study [17]. Inclusion for this qualitative phase were that women had participated in the FFB quantitative study and had been pregnant (with a single baby) and a booking-in BMI $\geq$35 kg/m$^2$ as per the inclusion criteria for FFB+ quantitative phase [17, 18]. At the time of data collection BMI $\geq$35 kg/m$^2$ was the cut-off for accessing more specialist antenatal care at the study hospital. Women were also invited to participate if they had consented to participate in FFB but had not received an appointment to attend the research clinic because maximum capacity had been reached.

Details of the participants are provided in Table 1, but in summary: Mean age was 29.6 years (range 21–40); mean BMI was 37.3 kg/m2 (range 35–51); four were expecting their first baby, and fifteen were White British, ethnicity, (one = Black African, one = Asian, one = Middle Eastern).

### Interview schedule

A semi-structured interview schedule included questions related to weight and lived experiences of managing weight, particularly during pregnancy, experiences of antenatal care, motivations for participating in the Fit for Birth study and awareness of community-based maternity services (see Table 2).

**Table 1. Participant characteristics.**

| Number ID | Age | BMI | Parity | Ethnicity |
|---|---|---|---|---|
| 1 | 35 | 40 | P | WBr[a] |
| 2 | 31 | 36 | M | WBr[a] |
| 3 | 27 | 43 | M | ASIAN |
| 4 | 21 | 46 | M | WBr[a] |
| 5 | 26 | 51 | M | WBr[a] |
| 6 | 32 | 40 | P | BlAFRICAN[b] |
| 7 | 30 | 38 | M | WBr[a] |
| 8 | 40 | 44 | M | WBr[a] |
| 10 | 34 | 35 | M | WBr[a] |
| 11 | 38 | 44 | M | WBr[a] |
| 12 | 25 | 46 | P | WBr[a] |
| 13 | 29 | 44 | M | M.EAST[c] |
| 14 | 27 | 39 | M | WBr[a] |
| 15 | 39 | 44 | M | WBr[a] |
| 16 | 26 | 43 | P | WBr[a] |
| 17 | 23 | 41 | M | WBr[a] |
| 18 | 23 | 47 | M | WBr[a] |

[a] WBr = White British

[b] BlAfrican = Black African

[c] M. EAST = Middle Eastern

## Procedure

During data collection for FFB, participants were asked to indicate if they were willing to be interviewed. Of 78 eligible women, 69 were subsequently contacted by the researcher by phone or letter. Nine women had left the area and were no longer contactable. Thirty-six women were unresponsive, and 15 changed their minds and declined to be interviewed, leaving 18 women. Participants signed a consent form, and a mutually convenient time and location for the interview was agreed. Most women were interviewed in their own homes, with one choosing to be interviewed in a private room at her place of work. All interviews began with the researcher introducing herself and explaining that the interview aimed to explore their perceptions and experiences of living with obesity during pregnancy, particularly with regards to antenatal care received.

The researcher made notes following the interviews and discussed her reflections during supervision (with JA and LN). This is an effective way of evaluating interview techniques and maintaining reflexivity [20].

## Data analysis

The interviews were recorded using digital voice recorders. These audio recordings were subject to verbatim transcription and thematic analysis was applied to this data [15, 16] (see

**Table 2. Sample of interview questions.**

| |
|---|
| "What were your experiences of antenatal care?" |
| "How did you feel when you were asked to participate in the 'Fit for Birth' study?" |
| "Were you aware of any community services available to pregnant women and did you participate in any of them?" |
| "Did you attempt to lose weight prior to becoming pregnant and if so via what methods?" |

**Table 3. Analytical procedure.**

| The application of step-by-step thematic analysis (Braun and Clarke 2006) | |
|---|---|
| 1. Familiarising yourself with your data | The interview recordings were transcribed verbatim (MC), following which, the authors (MC, JA, LN) read and reread the transcripts to become familiar with the breadth and depth of data discussed and initial ideas were noted. |
| 2. Generating initial codes | The authors (MC, JA, LN,) created initial codes systematically, on a line-by-line basis, pertinent to the research question. The codes were then collated across the whole data set. |
| 3. Searching for themes | Codes were collated into potential themes (by MC, JA, LN,) |
| 4. Reviewing themes | Discussion and generation of themes took place via face-to-face and online meetings with the authors. This helped to ensure that themes were relevant to the related coded abstracts and the entire data set (MC, JA, LN). The analytical strategy was data driven and inductive, with a focus on discussing and identifying the principal themes that repeated throughout transcripts. Themes were revised then validated across the data; quotations were chosen to illustrate identified themes. Lastly, a thematic map was generated (LN and JA). |
| 5. Defining and naming themes | Themes were defined and the overall story of the analysis was drafted (MC, JA, LN) |
| 6. Producing the report | The analysis was refined, linking the findings to previous literature and the research question. The broader impact of the findings was considered (MC, JA, LN). |

Note: Author initials in brackets to indicate task completed.

Table 3). Discussions between the authors took place to enhance the definition and refinement of themes until an agreement was reached.

The first and lead author (MC) was a registered nutritionist with a research interest in maternal nutrition and health. Second author (LN) is a Registered Practitioner Psychologist (Health) and a Reader in academia, with clinical experience in obesity services and research expertise in qualitative methodology. The third author (AW) was the PI for the original FFB study. The last author (JA) was a Registered Dietitian, with >12 years of clinical experience in maternity care and a Reader in academia. All authors are parents and had various experiences of encountering antenatal and postnatal care services. These personal experiences were deemed helpful to be able to empathise and understand the women's perspectives, though care was taken to be reflective and to not direct the analysis towards any specific narrative. As a multidisciplinary team, we recognised how our individual clinical, academic, and personal experiences could influence the analysis. We endeavoured to ensure that the discussions during the analysis promoted self-awareness and authentic interpretation of the data throughout [21].

## Results

Eighteen interviews took place, lasting between 30–60 minutes in duration. One recording was omitted from the analysis because the quality was very poor, resulting in 17 transcripts for full analysis. Four core themes were developed from the analysis:

1. Antenatal care is inconsistent

2. Additional support is needed for those with a high BMI in pregnancy
   2.1 Low engagement with community maternity services

3. Women feel judged because of their weight

4. Weight cycling is highly prevalent in women living with obesity

## Antenatal care is inconsistent

It is apparent from this research that women had differing experiences of antenatal care, ranging from very positive to very negative. Positive experiences included the feeling that they were more closely monitored in subsequent pregnancies. For example, if they developed gestational diabetes in an earlier pregnancy, they were more closely monitored during the subsequent pregnancy:

> "*I was monitored more under the hospital, and I was tested for the diabetes much earlier the second time and it came out that I did have it again and then again, I was under the care of the hospital each week*" [R2]

Most of the positive comments related to the midwives being *kind* or *nice*, and that the overall experiences were '*great*', '*a really good experience*' or that the HCPs that they encountered were '*really helpful*'. Reinforcing the notion of the special relationship that develops between pregnant women and midwives [22]. However, some women expressed disappointment that this special bond didn't develop due to a lack of consistency with midwives because of differing shift patterns.

> "*Like on the ward there was no consistency with midwives, and I realise that their shift pattern changes but there seemed to be no consistency*" [R8]

Some women felt that the care they received was more inconsistent and differed between HCPs with positive experiences with midwives and more negative experiences with the consultants. For example:

> "*It was absolutely great. . .couldn't have faulted it [*midwifery care*], erm, the only problem was one consultant that was a bit funny with me. . .when I requested to be induced, he was like 'no, you know it's not happening, you know you're overweight, that's your problem*" [R4]

## Additional support needed

Anxiety about living with obesity during pregnancy was reported by many women. For example, there was a suggestion by one woman that she was being ignored and unsupported in the second trimester. Despite living with obesity, she felt that HCPs mostly consider risk to be associated with the early stages of pregnancy:

> '*once you've passed your 1$^{st}$ trimester it's. . . well my community midwife. . .I didn't want to you know report her but. . .I'd sent her text messages and I called her and she used to switch off her phone and you know. . .and I would. . .I sent her a few text messages and she never replied you know and I said 'look this is really urgent and I'm really worried cos I'm having this wet. . .wet discharge' and she wouldn't reply and sometimes I would try calling her and she used to switch off her phone so I felt bad. . . I was just so traumatised, and I was just concentrating on the loss. . .trying to save my baby's life* [R6]

Most women agreed that having extra appointments, especially in the second trimester when appointments are typically less frequent, would be beneficial for them, for example:

> "*Once you get past the 5 months there's no scans. There is nothing in there unless they think there is something wrong*" [R5]

Some of these concerns were influenced by media stories regarding obesity in pregnancy, which was effectively summed by one woman who commented:

*"With everything you read in the media you know. . .you see and think 'oh I'm gonna get diabetes. . .I'm gonna die during labour cos I'm so huge' which were the worries I did have"* [R10]

However, it was not only a lack of appointments that women felt were in shortfall but a lack of continuity regarding contact with their midwife. Several factors affected contact with the midwife which included bad weather (affecting public transport), midwives not being available via text messaging services and a lack of follow-up support. This led to some women feeling like they were disregarded and unclear (*'in the dark'*) as to what was going on.

*"There were times when I felt that I needed more care, and I didn't get the care. . .so that got me upset a bit you know. . . just before I lost the baby. . .I think when I entered, I think about my 20th week. . .19 or 20 weeks that's when I started having problems and I felt I wasn't being listened to really"* [R6]

It is possible that because of the perceptions of inconsistency in care, many women actively sought participation in health research and welcomed the opportunity to participate in the Fit for Birth (FFB) research study. General perceptions of participating in the study were overwhelmingly positive. The women believed that recording their dietary intake and being weighed at each appointment helped them to monitor their own weight and encouraged more positive eating behaviours, for example:

*"I wanted to cos, I wanted the study so I could monitor my weight, so I don't put on too much weight, especially the diary you know. . .writing down what you eat that was really helpful it made me more conscious"* [R6]

There was also a consensus that the additional appointments helped to alleviate some of their anxieties and concerns, particularly in the mid-gestational phase.

*"at least if I take part in FFB I almost felt as if I would be better looked after through the process, `and it did feel as if I was really"* [R8]

## Engagement with community maternity services

In contrast to the perception that there is insufficient or inconsistent antenatal care, the women were asked if they were aware of or had participated in any pregnancy-related community services.

Very few women were informed of any community services available to pregnant women with a BMI of more than 30kg/m$^2$. Those that had been informed of services were advised of aqua natal and pregnancy yoga classes, both of which were open to all women regardless of BMI. When asked if they had engaged with the recommended services, it was apparent that none of the women had. Several reasons for this lack of engagement were cited, including work commitments, childcare difficulties, and access. However, other reasons given related to confidence issues, for example:

*'I'm not the sort of person who can go into these sorts of groups and everything's all lovely. . .I'm not that person to sit there and go "oh hi" you know and all this "and I'm overweight". . .it's not me'* [R4]

This demonstrates the perception of weight stigma that exists for women living with overweight and obesity within society. However, these examples are specific to experiences within healthcare settings.

## Women feel judged because of their weight

Women recounted unpleasant and judgemental comments made by various HCPs. One participant shared her experience of feeling unsupported after losing her baby in the second trimester due to pre-eclampsia:

*"the only problem I had was with one doctor after I'd lost the baby. . .I think he was talking a bit matter of fact you know like doctors do. . .they see it every day don't they and he wanted to put me on a ward and I was saying 'I'm not going on a ward with people who have just had baby's I'm just not doing it' and he was going 'the thing is like. . .you're fat. . .you're baby's dead'""[R16]*

Other women experienced more generalised weight stigma from their GP, reporting that GPs often implied that obesity underpinned all aspects of adverse health conditions:

*"I think the only thing is with the doctors and that. . .because they put everything down to the weight part of you and sometimes it's not and you know it's not. . .you know like I went one week, and I was breathing bad and I had a chest infection and he was sort of looking at me as if to say 'it's cos you're overweight" [R15]*

Even being asked to participate in the FFB study had a negative impact on some women, for example:

*"Well, it just basically made me feel 'oh, she's big, fat and pregnant'" [R1]*

The overall influence of obesity often led to anxiety and guilt in many of the women, who felt that their weight had negative impacts on their babies' health:

*"Anxious, cos obviously I'm putting my baby at risk because of my weight" [R5]*

Other participants felt that meeting the criteria for accessing fertility treatment was also judgemental:

*"Oh no you can't have IVF cos you're overweight. . .see I think that's wrong cos I don't think that should matter, it's down to whether you can carry a baby or not isn't it. . . it's not down to being overweight I don't think, not being able to fall pregnant. . .we're all as good as each other whether we be skinny or whether we be fat" [R11]*

Ultimately, the women themselves were the harshest judges of their weight, often using quite negative terminology to describe themselves and identifying certain foods as *'naughty'* or *'bad'* if they had *'fallen off the wagon'*. Conversely, they would describe themselves as being 'good' if they were following a dietary regimen and avoiding the foods more associated with an unhealthy dietary pattern.

## Weight cycling

Many of the women described their difficulties in losing weight and maintaining weight loss as a 'battle' or a 'vicious circle'. The term coined for this phenomenon is weight cycling [23].

All the women had previously attempted to lose weight. Many had entered a weight cycling pathway with weight loss followed by regain. This was particularly associated with life events, including pregnancy, followed by the difficulties of losing weight following pregnancy.

*"I did start putting weight on then I lost weight, then I got married and had my daughter, that's when I put more weight on"* [R3]

It was also acknowledged that a successful weight loss attempt was almost always followed by a period of regain which often led to an increase in the amount of weight gained.

*"My weight over the years has been up and down and I've dieted and lost weight then I've put it all straight back on again plus more"* [R12]

Women revealed that their weight often fluctuated with significant life events; their motivation to reduce weight often coincided with having children, getting married or key birthdays, for example:

*"Before my 21ˢᵗ birthday, I was at Weight Watchers, and I became a gold member"* [R7}

Their response to this heightened motivation was to sign up to a 'slimming group' or an exercise class at the gym, such as *boxing, Zumba, and swimming;* most experienced some success with this:

*"Before I got pregnant this time round, I did lose some weight with Weight Watchers"* [R6]

However, experiences at slimming clubs were often negative; women reported that "*they didn't like discussing weight in front of everyone";* with others describing a lack of discussion or support:

"*they're just there to weigh you and stuff. . .they're just there to get your money off you really, aren't they*?*" [R15]

Most women expressed that the motivation and success were often short-lived; their weight would "*creep back on*" eventually. Several women reported multiple attempts at achieving weight loss, often using more extreme regimes such as "*the stupid cabbage soup diet*" or tablets that "affect appetite" but none experienced any long-term success:

"*I've tried everything, everything, Weight Watchers, Slimming World, doctors, slimming tablets, gym. . .everything. . .but nothing seems to work"* [R4]

Some women reported achieving weight loss to improve their chances of conceiving, especially if they had experienced trouble with this:

*"I had tried for 4 years to have a baby, and it had never happened, so we were both concerned about our weight"* [R8]

Once children arrived, women often reported that their opportunities to exercise were reduced significantly, either due to time restrictions or cost:

*"I used to take the kids. . .they used to go in the kid's club. . .I'd go to the gym for an hour. It's so expensive. . .seven years ago, you only had to pay for yourself. . .but now you have to pay for them all"* [R11]

However, it is notable that all these women had a BMI over 35kg/m$^2$ at the time of participation in the FFB study, demonstrating the difficulties in maintaining a healthy weight despite continual efforts to achieve this.

## Discussion

This study identified key issues facing pregnant women with obesity regarding accessing maternity and support services during antenatal care. This research appears to be one of the few studies exploring the lived experiences of women recruited to participate in an observational study of obesity during pregnancy and to explore their perceptions of the healthcare they receive during this time. This research contributes insight to both the care experiences of women living with obesity during pregnancy, but also offers insights into the recruitment and research processes for women living with obesity and invited into clinical research trials. In line with other relevant studies [24], many women reported positive aspects about the antenatal care that they had received. However, there were several areas where the women reported negative experiences, and recommendations for improvements in service delivery are provided.

### Women feel judged because of their weight

Perceptions of weight bias in HCP, towards the general patient population living with overweight and obesity, have been reported in a number of UK studies [25–27]. Similarly, pregnant women living with obesity have also experienced weight stigma from HCPs.

Reports of rude and disrespectful healthcare staff that lack compassion and empathy [26] were mirrored by the pregnant women who participated in this study, with particular reference made to consultants. Weight stigma has been shown to reduce engagement with positive health-related behaviours and is associated with increased food consumption and episodes of binge eating [28]. These experiences are viewed negatively and potentially lead to further detrimental effects on mental and physical health. Furthermore, some of the terminology used to describe obesity has been shown to impact women's emotional responses to HCPs. Previous research has discussed how terms such as 'superobese', 'extra-large' or 'fat' are negatively perceived by patients who had a preference for terms such as overweight, unhealthy weight, or preferred being described according to body mass index (BMI) [29, 30]. In this study, some of the women described events in which they were referred to by negative terms, such as fat. When explaining the risks associated with gestational weight gain, previous research has reported how women would prefer to be approached with understanding. Women want to be treated with respect, rather than being offered unsolicited advice or having weight-biased assumptions about food or exercise made [31]. It is noteworthy that women often find the conversations around weight to be both disrespectful and unhelpful. Other researchers [27] indicate that a paradigm shift is needed regarding the blame culture and individual responsibility around obesity. People living with obesity, (including pregnant women), demonstrated a reluctance to discuss aspects of their weight with HCPs, and equally, HCPs expressed discomfort at having to raise issues around weight with patients. This is partially due to uncertainties regarding the referral of women to specialist services to manage weight during pregnancy [8]. In this study, the women reported being less likely to engage with weight management services if they had had a negative experience regarding a lack of, or a poor explanation of why the

referral to weight management support was necessary'; similar findings have been reported elsewhere [13]. This demonstrates that how HCPs communicate with women about their weight is never neutral, having either a positive or a negative effect depending on the conversation. This lack of neutrality, based on Freire's philosophy, has been reported elsewhere regarding health education [32]. A 2022 review suggests that reducing weight bias in health care should be a priority and although evidence indicates conversations around reducing weight bias in HCP are encouraging, more interventions and training strategies are needed. A move away from weight centred health care approaches to more health related approaches is recommended [33].

As suggested by the women in this study, the HCPs (including midwives) who have contact with pregnant women living with obesity, need training regarding weight stigma and the negative impact certain words and phrases can have on individuals in their care. Previous research [34] suggests that HCPs need to become aware of their unconscious bias regarding weight before efforts can be made to improve communication and behaviour towards patients living with obesity. They also reported that HCPs currently receive very little training regarding obesity and weight bias [34]. Midwives are often reluctant to broach the subject of obesity with pregnant women for fear of offending [8, 35]. However, due to the increased numbers of women presenting for antenatal care with a BMI more than 30kg/m$^2$, it is suggested that obesity has been normalised in pregnancy by midwives and some of the issues associated with higher BMIs in pregnancy are not raised [8]. As a result of not acknowledging higher BMI during consultations, weight management or healthy eating advice is often not offered [8]. Furthermore, most UK antenatal clinics only have sufficient resources to offer specialist support to pregnant women with BMI > 40 kg/m$^2$, meaning that women with a BMI 30–39.9 kg/m$^2$ can only access routine care despite having a higher risk. This is not unique to the UK with reports of similar issues occurring globally, for example in Brazil, midwives receive comprehensive training with regards to nutrition and weight management, however, pregnant women have very limited access to midwives, (possibly due to there being only one undergraduate midwifery training programme available for the entire population) [17]. It is important that antenatal consultations with midwives offer information and support regarding diet and weight, as this may be the only opportunity to discuss these issues during the entire pregnancy journey.

## Antenatal care is Inconsistent

Antenatal care in the UK follows the midwife-led model of care, where the organisation of care from booking in to delivery is built around a partnership between midwife and women [36]. This care is based on a supportive and trusting relationship between midwives and the women in their care [37]. Many women in this study reported positive experiences of their antenatal care, but many also reported more negative aspects of care. Research studies have reported delays or avoidance of appointments or interactions with healthcare services due to weight stigma where HCPs have demonstrate weight bias, leading to inconsistencies in care [34]. A key factor that may contribute to inconsistencies in the care of pregnant women with obesity is the lack of UK guidelines regarding healthy GWG. Previous research has found that midwives were reluctant to offer advice about GWG, citing a lack of UK guidelines as a reason for this [8].

Moreover, when asked what GWG they would recommend, a wide range of figures were suggested, ranging from 12 to 20kg. Additionally, only a quarter (25.4%) of pregnant women have reported receiving *any* advice about weight and concluded that it was unclear and lacked detail when it was given [38]. Furthermore, information about diet and exercise was described as 'brief' and led to women seeking advice from unreliable sources (such as the internet/social media) [38].

Previous research has highlighted some of the negative stereotypical beliefs for people living with obesity held by some HCPs, including terms such as 'stupid', 'unsuccessful' and 'lazy' being reported [25]. These negative perceptions of people with higher BMIs related to HCPs perceptions of poor self-management behaviours [39]. However, it was determined that knowledge regarding the aetiology and risk factors associated with obesity were relatively low in some HCPs with the belief that obesity is in the control of the individual still persisting [25, 40]. Targeted educational strategies regarding the uncontrollable factors associated with the onset of obesity and the negative impact on mental health among patients living with obesity may go some way to improving the HCP and patient relationship [25]. The 'Care and decision making in pregnancy' document (2022) by NIHR [41] points out that 'midwife-led continuity of care' not only saved babies' lives but also improved women's overall experience and enhanced pregnancy outcomes. To support this approach, it's essential that women have access to clear information for making informed decisions about their care. Furthermore, current UK guidelines (NICE 2021) [42], expect midwives to have conversations with ALL pregnant women about diet and weight; also, we know that pregnant women want this information from midwives as part of their package of care [14]. A key aspect of this model involves training midwives to provide comprehensive diet and weight management advice to all pregnant women, regardless of their BMI, aiming to minimise variations in care.

## Additional support needed

Many of the women who participated in this study perceived that living with obesity exacerbated some pre-existing conditions, such as asthma, sciatica or was implicated in the development of gestational diabetes mellitus (GDM) [43]. Evidence suggests that women with obesity are at risk of pregnancy complications such as pre-eclampsia, which is most likely to occur after 20 weeks' gestation and can be life-threatening to both mother and infant [44].

Other women expressed fears that living with obesity may compromise the health of the foetus, leading to further fears and anxieties. As such, although women expressed dissatisfaction with the consistency of antenatal care, paradoxically, they also wanted additional monitoring. There was a general sense that routine antenatal care was somewhat lacking following the antenatal booking-in appointment. The frequency of routine antenatal appointments increased in the third trimester. However, appointments were less frequent in the latter half of the second trimester, and the number of appointments varied depending on parity, with parous women potentially experiencing an eight-week gap between week 20 and 28-weeks' gestation. Nulliparous women often have additional appointments, including one at 25 weeks' gestation [6].Women with a booking-in BMI $> 40$ kg/m$^2$ are usually referred for additional support in pregnancy and may be seen more regularly [17]. For women with a BMI 30–39.9 kg/m$^2$, routine care may result in periods without follow-up by HCPs despite increased health risks, and these women should be fully informed about the risks of obesity in pregnancy [5]. However, many women feel unsupported if there is a lack of follow-up to monitor those risks.

Many women agreed to participate in the FFB study purely because they would receive additional appointments via the research clinics and what they perceived as enhanced care. This finding is supported by other studies [45]. Women may be motivated by many factors regarding participating in research; altruism and a desire to improve medical science and help others were cited as primary reasons. However, enhanced care and a perceived benefit to their health or foetal health were also cited as motivating factors, for this study [45]. Further to this, this study offers an insight into how recruitment processes and language can be used for future clinical research trials.

Regarding physical activity, the UK government recommends that women be physically active at a moderate intensity for approximately 150 minutes per week and include two sessions of muscle-strengthening activity [46]. This recommendation also applies to women not used to exercising, where the advice is to start gradually. The benefits of exercise during pregnancy are overwhelmingly positive, particularly in reducing gestational weight gain and incidence of gestational diabetes mellitus [47]. However, most women interviewed in this study, reported they had received very little advice from their midwives promoting any form of physical activity or recommendations to physical activity initiatives in the local area that could support them during pregnancy. The exceptions were when aqua natal and pregnancy yoga recommendations were provided, but uptake of these was very low, with most women reporting time constraints as a barrier to participation. Barriers to participation related to work or family commitments and either a lack of variety in the number of classes being offered or the inconvenience of the times they were available [48].

## Weight cycling

Participants reported being motivated to improve their health and/or weight by key life events such as pregnancy, weddings, and birthdays, but that motivation was often short-lived and a lack of time, work and family commitments were cited as barriers to maintaining diet and exercise regimes. This concurs with findings reported elsewhere [49]. So, although pregnancy is often cited as a 'teachable moment' for lifestyle change [50], future interventions need to consider how to maintain motivation in the longer term.

## Limitations

Although more recent evidence [8] demonstrates that midwives still have difficulties in raising sensitive subjects with pregnant women living with obesity, a limitation of this study is that the data was collected over 10 years ago. Therefore, training relating to the treatment of individuals living with obesity may have been introduced to reduce weight bias.

## Conclusion

The role that midwives and other HCPs play in the care of pregnant women is essential for positive pregnancy outcomes. However, despite good intentions, it was evident that there were still considerable inconsistencies in the standard and continuity of care for pregnant women, particularly those living with obesity at the time of data collection. Furthermore, pregnant women living with obesity were often left feeling confused, uncomfortable, overlooked, and discriminated against; with reports of weight bias still being experienced. However, ignoring the issues and avoiding such discussions appear to only cause further anxiety for pregnant women, who understood their pregnancy to be higher risk. A lack of follow-up during trimester two in this study was identified as a concern. There is an urgent need to ensure that pregnant women living with obesity have access to enhanced care, which reflects their risk. Moreover, multidisciplinary training for HCPs is also needed, which addresses the concerns of those living with obesity and seeks to improve healthcare delivery, based on some of the experiences of pregnant women, living with obesity, as outlined in this study'.

## Author Contributions

**Conceptualization:** Margaret Charnley, Julie Abayomi.

**Data curation:** Margaret Charnley.

**Formal analysis:** Margaret Charnley, Lisa Newson, Julie Abayomi.

**Funding acquisition:** Andrew Weeks, Julie Abayomi.

**Investigation:** Margaret Charnley, Julie Abayomi.

**Methodology:** Margaret Charnley, Julie Abayomi.

**Project administration:** Julie Abayomi.

**Supervision:** Julie Abayomi.

**Visualization:** Margaret Charnley, Lisa Newson, Julie Abayomi.

**Writing – original draft:** Margaret Charnley, Lisa Newson, Julie Abayomi.

**Writing – review & editing:** Margaret Charnley, Lisa Newson, Andrew Weeks, Julie Abayomi.

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
