## [Decision Letter · Decision Letter 0]

22 Jan 2024

PONE-D-23-39200A qualitative exploration of the experiences of pregnant women living with obesity and accessing antenatal carePLOS ONE

Dear Dr. Abayomi,

Thank you for submitting your manuscript to PLOS ONE. After careful consideration, we feel that it has merit but does not fully meet PLOS ONE’s publication criteria as it currently stands. Therefore, we invite you to submit a revised version of the manuscript that addresses the points raised during the review process.

We look forward to receiving your revised manuscript.

Kind regards,

Fekede Asefa Kumsa, PhD

Academic Editor

PLOS ONE

Journal Requirements:

"MC - MerseyBEAT, Liverpool Primary Care Trust; Liverpool Health Inequalities Research Group (Ref MBRP005)."

Reviewers' comments:

Reviewer's Responses to Questions

**Comments to the Author**

1. Is the manuscript technically sound, and do the data support the conclusions?

Reviewer #1: Yes

Reviewer #2: Yes

2. Has the statistical analysis been performed appropriately and rigorously? 

Reviewer #1: N/A

Reviewer #2: N/A

3. Have the authors made all data underlying the findings in their manuscript fully available?

Reviewer #1: No

Reviewer #2: No

4. Is the manuscript presented in an intelligible fashion and written in standard English?

Reviewer #1: Yes

Reviewer #2: Yes

5. Review Comments to the Author

Reviewer #1: Summary of the research and my overall impression

Thank you for giving me the opportunity to read and review this interesting and important manuscript about how pregnant women with obesity have experienced the antenatal care.

The study is important as weight related health in pregnancy affect generations, while existing interventions seem insufficient. The authors present findings that confirm earlier study results but also describes new angles, such as how willingness to enroll in a study may be driven by a perceived lack of existing support.

The method is described in a clear fashion and the writers present and discuss their findings in a relevant way. The quotes are well-chosen and support the results. I enjoyed reading it and believe it will be ready for publication after dealing with some minor, but important, issues and language revisions.

Areas of improvement

In the discussion section, I suggest the authors mention the limitation that the interviews were performed more than 10 years ago and thus pregnant women with obesity may have different experiences today, as the treatment and attitudes towards obesity may have changed.

For the same reason, I suggest that the conclusion section is slightly adjusted overall by using past tense to indicate that the conclusions are based on how women experienced the care over 10 years ago. For example, line 607-608, where it says ”…it is evident that there is considerable…” could rather read “…it is evident that there was considerable…”.

Language

The language is well-written. However, throughout the entire manuscript, including in tables, there are about 95 errors where space between words is lacking, such as “isimportant” (line 104) and “asinevitable” (line105) and so on. There are also a large number of missing spaces after commas, full stops and references such as in line 80 "collegues10found" or in line 81 “(n=92)reported” and “care,with”. The need for corrections should be easily found by using the grammar and spelling function in Word.

Besides these spelling error I suggest the following language edits:

Line 39 (introduction).

Remove either “a” or “the” in the first sentence.

Line 53

Are women really advised to lose weight intrapartum, as in during labour? If not please clarify or perhaps remove the word intrapartum.

Line 92

This sentence left me confused. Is there perhaps something missing at the end? Something like: “…if they did not attend the weight management course”. Please clarify.

Line 147, Table 1.

I wonder about the purpose of pseudonyms in the table. Unless they are used instead of numbers under the quotes, the column could be removed.

Line 450-453 (discussion section).

The sentence left me confused. It starts with “similar to” and continues with “but in contrast”. If the words “but in contrast” is removed it seems to make sense but I am not sure that is what the authors want to point out. Please clarify the meaning.

Line 491-492 (discussion section)

The statement “Midwives are often reluctant…” could use a reference, for example:

Shame and avoidance as barriers in midwives' communication about body weight with pregnant women: A qualitative interview study. Christenson, A., Johansson, E., Reynisdottir, S., Torgerson, J., & Hemmingsson, E. (2018). Midwifery, 63, 1–7. https://doi.org/10.1016/j.midw.2018.04.020

Other comment

I really liked that the authors had put examples of interview questions in a table (instead of me having to look for them in a supplementary material).

Reviewer #2: Thank you for the opportunity to review ‘A qualitative exploration of the experiences of pregnant women living with obesity and accessing antenatal care’. This paper provides an interesting insight into the antenatal experience of women with high BMI. Overall the paper is well written and provides useful recommendations for future interventions. The decision to omit strengths and weaknesses of the study in the discussion is regrettable, as it should be noted that the particular demographic of women interviewed in the study had already volunteered for a study on gestational weight gain, so observations and recommendations may be skewed by this association, which is not acknowledged in the paper. It is not clear why women with BMI >35kg/m2 were included when reference to either BMI >30kg/m2 or BMI >40kg/m2 are noted in other parts of the paper as having more significant influence on care provision. Further clarity is also needed on the participant group, whether they were pregnant at the time of interview or postnatal, and if interviews were postnatal how many weeks post pregnancy were the women? Please also provide consistent figures in how many were recruited/lost to follow up etc. Attention should be paid to grammatical and structural errors in this paper which are noted below. Overall, I believe this paper makes a valuable contribution to the literature on this topic, however the following concerns should be addressed prior to publication:

Abstract: Discussion of midwives beliefs and perceptions does not seem justified in the abstract as this paper is only exploring experiences of women.

Sentence "This study explored the experiences of accessing antenatal care in pregnant women living with obesity." - antenatal pregnant care in pregnant women does not make sense

Line 23 Repeated issue throughout this paper with words joined together. In this instance, foradvising and risksthroughout

Line 39 Pregnancy is often a the time – remove a

Line 46 Growing emphasis on not for

Line 53 Weight loss is not recommended intrapartum in the guidelines referenced. Provide supporting evidence if suggesting weight loss during pregnancy is advised.

Line 60 Reference 7 is exploring mothers’ experience of discussions on weight, not midwives experience, it does not seem an appropriate reference for a statement regarding midwives attitudes to weight counselling. There are a number of empirical sources to refer to on this topic.

Line 82 Source of weight stigma not cause

Line 87 Review this sentence for clarity

Line 91 What makes this more worrying?

Line 123 I would encourage use of primary references such as Braun and Clarke, rather than secondary references to justify this approach to data analysis

Line 124 Up to how many weeks postnatally were women included?

Line 128 Review this sentence for clarity

Line 141 Were all women pregnant or were some postnatal? If including postnatal women, what time period post-birth were these interviews conducted?

Line 141 Why was BMI >35kg/m2 chosen when you have noted classification of obesity and recommendations regarding obesity start at BMI 30kg/m2?

Line 161 Why were 9 women excluded from recruitment?

Line 164 78 – 9 – 35 – 15 = 19 or 69-9-35-15 = 10. The figures do not add up. Please review.

Line 178 Clarity required regarding approach to thematic analysis. Was it Braun and Clarke’s approach to Thematic Analysis as suggested in the table? Though data may have been analysed prior to Braun and Clarke’s 2022 Thematic Analysis textbook being printed, it may be worth clarifying explicitly that you used the 2006 approach and why you opted for that instead of the Reflexive Thematic Analysis of 2022 as this paper will be published >2 years after publication of the new Braun and Clarke approach.

Line 183 Full stop rather than semi-colon more appropriate here

Line 182 Review this sentence for clarity

Line 187 The word experience is used repeatedly in this sentence, perhaps consider alternative word?

Line 230 If this is a study of antenatal care, why is there reference to the ward? Was it an antenatal ward?

Line 242 Spelling error of ‘you’re’

Line 262 Full stop after them should be removed

Line 274 More context needed for why bad weather would impede contact with a midwife?

Line 276 Is ‘in the dark’ a quote from a participant?

Line 317 Are aquanatal classes provided within healthcare settings? If not, how are these examples specific to experiences within healthcare settings?

Line 358 Review use of comma alongside full stop and lack of capitalisation

Line 398 Which participant is this quoted from?

Line 432 Reconsider use of ‘one’ in this sentence as healthy weight is a range, not a singular figure

Line 441 contribute should be contributes

Line 445 Severe? Or several?

Line 450 This sentence does not make sense

Line 460 See also the systematic review by Puhl (2020) regarding weight related terminology, note that the term obesity, with or without person-first language, is widely considered stigmatising by the affected group

Line 465 Worth noting that in the field of Fat Studies fat is not considered pejorative and has been reclaimed, however agree that it was used negatively in these specific interactions

Line 472 Review this sentence for clarity

Line 476 Review this sentence for clarity ‘why the referral was to weight management support was necessary’

Line 477 A negative experience described by whom? Or is it that after women had a negative interaction they were reluctant to return?

Line 481 It is ‘never’ neutral? Is this supported by evidence?

Line 485 Do HCP have any training regarding weight stigma? Let alone ongoing?

Line 491 What training currently exists?

Line 492 Support this statement with evidence

Line 497 ‘Is’ should be added to this sentence - ‘is not offered’

Line 503 There is an undergraduate training programme for pregnant women?

Line 514 Many reported positive experiences and many reported negative, can you be more specific is there a figure? Did women who reported negative experiences also report some positive experiences or was it definitively one way or the other?

Line 516 Is this sentence referring to women in the study or patients more generally? It’s confusing to go from talking about participants in the study to patients generally without qualification.

Line 524 Of should be added to this sentence– ‘only a quarter of’

Line 530 Negative stereotypes lead to perceptions, consider rewriting this sentence to aid clarity in why terms such as lazy are reported – are these negative perceptions of people with high BMI or are these terms used in consultation…

Line 533 Knowledge of what? Knowledge of obesity in what context? Clinically?

Line 536 Reference needed for sentence starting with ‘The belief..’

Line 545 I encourage review of Fair & Soltani (2021) Meta review of lifestyle interventions to reduce gestational weight gain in this context. If encouraging midwives to provide comprehensive diet and weight management advice, is there clear evidence of benefit to maternal and/or fetal outcomes?

Line 545 Consideration needs to given as to whether UK or US spelling will be used in this paper. Both conventions noted, including behaviour and minimized. Opt for one convention and ensure consistency.

Line 548 How were women able to differentiate between pregnancy or obesity exacerbating asthma and sciatica?

Line 550 Evidence suggests rather than Indications are. Alternatively, what indications?

Line 588 Review use of commas in this sentence

Line 591 Consider use of word variability in this context

Line 596 And/or

Line 611 Is it possible to make this statement given no midwives were interviewed as part of this study?

Line 618 Review this sentence for clarity

6. PLOS authors have the option to publish the peer review history of their article (what does this mean?). If published, this will include your full peer review and any attached files.

Reviewer #1: No

Reviewer #2: No

---

## [Author Response · Author response to Decision Letter 0]

14 Mar 2024

Thank you for the suggested amendments to our paper, we believe we have addressed each of them now (see attached table)

---

## [Decision Letter · Decision Letter 1]

9 Apr 2024

A qualitative exploration of the experiences of pregnant women living with obesity and accessing antenatal care

PONE-D-23-39200R1

Dear Dr. Abayomi,

We’re pleased to inform you that your manuscript has been judged scientifically suitable for publication and will be formally accepted for publication once it meets all outstanding technical requirements.

Kind regards,

Fekede Asefa Kumsa, PhD

Academic Editor

PLOS ONE

Reviewers' comments:

Reviewer's Responses to Questions

**Comments to the Author**

1. If the authors have adequately addressed your comments raised in a previous round of review and you feel that this manuscript is now acceptable for publication, you may indicate that here to bypass the “Comments to the Author” section, enter your conflict of interest statement in the “Confidential to Editor” section, and submit your "Accept" recommendation.

Reviewer #1: All comments have been addressed

Reviewer #2: All comments have been addressed

2. Is the manuscript technically sound, and do the data support the conclusions?

Reviewer #1: Yes

Reviewer #2: Yes

3. Has the statistical analysis been performed appropriately and rigorously? 

Reviewer #1: N/A

Reviewer #2: N/A

4. Have the authors made all data underlying the findings in their manuscript fully available?

Reviewer #1: Yes

Reviewer #2: No

5. Is the manuscript presented in an intelligible fashion and written in standard English?

Reviewer #1: Yes

Reviewer #2: Yes

6. Review Comments to the Author

Reviewer #1: Dear authors,

I consider your amendments to be satisfactory and after making two further minimal changes (which does not need another round of review) I believe the article is ready for publication.

The changes are:

Line 391

To be consistent throughout your quotes, you should remove the full stop after the quotation mark and before [R6]. It now says: Watchers”. [R6] while the other quotations do not have a full stop there.

Line 511 Please remove the apostrophe after the word “care”

Reviewer #2: (No Response)

7. PLOS authors have the option to publish the peer review history of their article (what does this mean?). If published, this will include your full peer review and any attached files.

Reviewer #1: No

Reviewer #2: No
